# Acoustoelectric Effect for Rayleigh Wave in ZnO Produced by an Inhomogeneous In-Depth Electrical Conductivity Profile

**DOI:** 10.3390/s23031422

**Published:** 2023-01-27

**Authors:** Cinzia Caliendo

**Affiliations:** Institute for Photonics and Nanotechnology, IFN-CNR, Via del Fosso del Cavaliere 100, 00133 Rome, Italy; cinzia.caliendo@cnr.it

**Keywords:** acoustoelectric effect, electrical conductivity, Rayleigh wave, phase velocity, propagation loss

## Abstract

The acousto-electric (AE) effect associated with the propagation of the Rayleigh wave in ZnO half-space was theoretically investigated by studying the changes in wave velocity and propagation loss induced by in-depth inhomogeneous changes in the ZnO electrical conductivity. An exponentially decaying profile for the electrical conductivity was attributed to the ZnO half-space, for some values of the exponential decay constant (from 100 to 500 nm), in order to simulate the photoconductivity effect induced by ultra-violet illumination. The calculated Rayleigh wave velocity and attenuation vs. ZnO conductivity curves have the form of a double-relaxation response as opposed to the single-relaxation response which characterizes the well-known AE effect due to surface conductivity changes onto piezoelectric media. As to the author’s knowledge, this is the first time the double-relaxation AE effect has been theoretically predicted.

## 1. Introduction

Rayleigh waves are surface acoustic waves (SAWs) traveling along the surface of piezoelectric half-spaces. Most of the wave energy is trapped at the surface of the propagating medium up to about one wavelength in depth: the amplitudes of both the mechanical and electric field wave components decrease exponentially in the bulk. By patterning interdigitated transducers (IDTs) onto the piezoelectric material’s surface and applying an RF signal, the IDTs excite the piezoelectric material to generate propagating SAWs whose frequency, amplitude, and wave-front orientation are defined by the dimensions of the electrodes, phase velocity of the material, and input power of the applied RF signal. The changes in velocity and propagation losses of a SAW due to the presence of a conductive thin layer covering the surface of the propagating path are known as the acoustoelectric (AE) effect: as changes in the conductivity of the layer may strongly affect the SAW propagation, the AE effect finds useful applications in the field of SAW-based sensors operating in both gaseous and liquid environments [1]. In reference [2], a hydrogen sensor is described which is based on the AE effect: a thin Pd film covering the SAW propagation path in a yz-LiNbO_3_ substrate adsorbs hydrogen, thus varying its electrical conductivity. The changed wave velocity and propagation loss represent the SAW sensor response to different hydrogen concentrations. In reference [3], a shear horizontal acoustic plate mode (SHAPM) sensor is studied, which measures the electrical conductivity of a liquid environment contacting one side of the quartz plate. The electric field associated with the acoustic wave extends several micrometers into the liquid and interacts with ions and dipoles in solution. Changes in the conductivity of the solution perturb the propagation velocity of the plate mode. In reference [4], the SHAPM devices are studied for sensing potassium ion concentrations in water. In reference [5], the SAW velocity and attenuation are measured as a nickel film is deposited onto a quartz SAW device: the AE response causes a rapid drop in velocity and a peak in attenuation over the 10 to 30 Å thickness range. The valley AE effect is described in reference [6]: the electric current densities in LiNbO_3_/dielectric/MoS_2_ were calculated and their magnitudes were compared with the conventional diffusive current.

The cited examples have a common term: the conductivity changes take place on the surface of the piezoelectric SAW propagating medium. ZnO is a piezoelectric wide-band-gap semiconductor material [7,8]: if illuminated by UV radiation, free carriers are photogenerated, resulting in a non-uniform conductivity change across its depth; it seems realistic to assume that the ZnO conductivity depth profile follows that of the UV energy density absorbed. The present paper studies the AE effect due to an exponentially decaying conductivity change across the depth of the ZnO substrate, such as occurs in the case of ultraviolet (UV) light absorption phenomena. Since ZnO can perform both functions together, SAW transductor and photoconductor, a careful study of the UV sensing device performances is required as a function of some design parameters, such as the acoustic wave type or the UV penetration depth, to cite just a few. 

The calculated data reveal that the velocity and propagation loss vs. conductivity curves have the form of a double-relaxation response as opposed to the single-relaxation response characterizing the well-known AE effect due to surface conductivity changes onto piezoelectric media.

To the best of the author’s knowledge, the double-relaxation AE effect has not yet been predicted theoretically, and no experimental results have been published yet; it is the author’s opinion that this phenomenon may affect the responses of the UV SAW sensors, and that much still needs to be further studied in the field. 

## 2. The AE Effect

Piezoelectric materials have an interesting feature: when subjected to a mechanical stress, they polarize and generate a bias voltage (direct piezoelectric effect); conversely, they undergo a mechanical stress when subjected to an electric field (inverse piezoelectric effect). The coupling between electric field and strain in piezoelectric solids is accounted for by the piezoelectric constitutive relations [9], which describe the interplay of stress *T*, strain *S*, and electric field *E* or potential Ф:(1)$$T_{ij}=c_{ijkl}^{E}\cdot S_{kl}-e_{kij}E_{k\;}$$
(2)$$D_{i}=\varepsilon_{ik}^{S}\cdot E_{k}+e_{ikl}\cdot S_{kl}$$
where *T_ij_* represents the stress vector (N/m^2^), *c_ijkl_* the elastic stiffness matrix (N/m^2^), *e_ijk_* the piezoelectric stress matrix (C/m^2^), *ε_ij_* the permittivity matrix (F/m), Ek=−∂Ф∂xk the electric field vector (V/m), $S_{kl}=\frac{1}{2}\left(\frac{\partial u_{k}}{\partial x_{l}}+\frac{\partial u_{l}}{\partial x_{k}}\right)$ the strain component, $u_{k}$ the mechanical displacement component along the Cartesian axis $x_{k}$ (x_1_ = *x*, x_2_ = *y*, x_3_ = *z*), and *D_i_* is the electrical displacement (C/m^2^). The superscripts *E* and *S* denote that the constants are evaluated at a constant electric field and strain, respectively.

The quasi-static equations for modeling the surface acoustic wave propagation in a piezoelectric elastic medium include the equations of motion [9].
(3)$$\frac{\partial T_{ij}}{\partial x_{i}}=\rho \cdot \frac{\partial^{2}u_{j}}{\partial t^{2}},$$
and the charge conservation law [9]:
(4)$$\frac{\partial D_{i}}{\partial x_{i}}=0$$

By substituting the stress–charge constitutive relations in Equations (3) and (4), a set of four equations is obtained, whose time-harmonic solutions are assumed as wave propagating along the *x*-direction [9]:(5)$$u_{i}=\beta_{i}e^{jkbz}e^{jk\left(x-vt\right)}$$
(6)$$\phi =\beta_{4}e^{jkbz}e^{jk\left(x-vt\right)}$$
in which *k* is the wave number, $j=\sqrt{-1}$, *z* the vertical direction, *v* and *t* the velocity and time, respectively; *β_i_* (for i = 1, 2, 3) and *β*_4_ are the amplitudes of the particle displacement components and of the electric potential. The boundary conditions for SAWs traveling along the surface of the half-space (*z* = 0 is the surface and the *z* axis points toward the bulk) are the following: 1. the normal components of the stress tensor $T_{3i}=0\;$must be zero at the free surface (*z* = 0); 2. the displacement components and electric potential must vanish at large depths (*z* →∞); 3. continuity of the potential and of the normal component of the electric displacement across the free surface (*z* = 0) of the piezoelectric half-space ($D_{3}=$ $D_{3}^{air}$ and Ф = Ф^air^ for *z* = 0). Details on the theoretical aspects of acoustic wave propagation in solid media are discussed in references [9,10].

Rayleigh waves are surface acoustic waves that travel in semi-infinite solids with a thickness much greater than the acoustic wavelength λ: the motion is confined to the near-surface region, while decaying below the surface, so that the whole wave energy is almost confined within a depth of about one wavelength. If the propagating medium is piezoelectric, a layer of bound charges is generated at the surface, which generates a wave potential traveling in synchrony with the mechanical wave. The mechanical wave and the wave potential are coupled together so that any surface electrical perturbation results in a change in wave velocity and propagation loss. Such perturbation can arise from the presence of a thin conductive layer covering the surface of the piezoelectric medium. The conductivity changes in the thin surface film affect the SAW/charge-carrier coupling: the resulting effect, the SAW perturbed propagation characteristics, is called the acousto-electric (AE) effect. The propagation of a SAW on a piezoelectric semiconductor with homogeneous bulk conductivity and covered by an ultra-thin conductive layer has been analyzed in reference [11], this problem having been previously addressed by Ingebrlgtsen [12], Kino and Reeder [13], Adler [14], and Datta [15]; the following formulas summarize the essential results, the equations that describe the changes in SAW velocity Δ*v* and attenuation *α* [16]:(7)$$\frac{\Delta v}{v_{0}}=-\frac{K^{2}}{2}\frac{\sigma_{s}^{2}}{\sigma_{s}^{2}+{\left[v_{0}\left(\varepsilon_{0}+\varepsilon_{s}\right)\right]}^{2}}$$
(8)$$\alpha =\frac{K^{2}}{2}\frac{kv_{0}\left(\varepsilon_{0}+\varepsilon_{s}\right)\cdot \sigma_{s}}{\sigma_{s}^{2}+{\left[v_{0}\left(\varepsilon_{0}+\varepsilon_{s}\right)\right]}^{2}}$$
being σ_s_ = σ · *h* the sheet conductivity, *h* the thickness, σ the bulk conductivity (the running parameter) of the conductive over-layer covering the piezoelectric half-space; K^2^ is the electroacoustic coupling coefficient of the piezoelectric half-space (about 0.96% for ZnO), *ε*_0_ and *ε_s_* the air and half-space dielectric permittivity, *k* = 2π/λ is the wavenumber, $\Delta v=v-v_{0}$, $v_{0}$ is the velocity of the SAW traveling along the bare surface of the piezoelectric half-space, v is the SAW velocity perturbed by the layer conductivity change, σ_c_ = $v_{0}\left(\varepsilon_{0}+\varepsilon_{s}\right)\;$is the critical conductivity corresponding to the attenuation peak. 

### Simulation Methodology and Results

A 2D FEM study was performed by using Comsol Multiphysics 5.6 software in the eigenfrequency study to evaluate the SAW velocity changes of the Rayleigh wave in the ZnO half-space covered by a thin Al layer. The model uses a piezoelectric multiphysics coupling node with solid mechanics and electrostatics interfaces. Figure 1 shows a schematic of a SAW delay line (consisting of two IDTs located onto the surface of a piezoelectric ZnO half-space and a thin metal layer covering the wave propagation path in between the transducers) together with the detail of the unit cell used for Comsol simulations. The cell, one wavelength (λ = 10 µm) wide, includes an insulating ZnO domain (7λ thick) covered by a very thin Al layer (10 nm thick), and by the air domain with a height equal to 2λ. Both mechanic and electrical fields were considered for the ZnO and Al layers (only the electrical field for the air domain); a free boundary condition was assumed for the bottom side of the ZnO substrate. The Al is assumed to have complex permittivity whose imaginary part is equal to $j\cdot \frac{\sigma }{\omega \varepsilon_{0}}$ where $\omega =2\pi f_{0}$, $f_{0}$ is the resonant frequency, and σ the Al electrical conductivity (the sweep parameter). It is assumed that x is the wave propagation direction and *y* is the half-space depth. Periodic boundary conditions are applied to the left and right boundaries of the unit cell, so reflections caused by the free edges can be ignored. Fine meshes (automatically generated, physics-defined triangular elements) were chosen for the FEM simulations.

A 2D eigenfrequency study was performed to calculate the Rayleigh resonant frequency (*f*_0_ = 253.5 MHz); a sweep parameter study was performed to calculate the real and imaginary parts of the mode eigenfrequency $f_{\sigma }$ at different Al layer electrical conductivity. Figure 2a shows the phase velocity $v=Real\left(f_{\sigma }\right)\cdot \lambda$ and propagation loss $\alpha =40\pi \cdot log_{10}e\cdot \frac{Imag\left(f_{\sigma }\right)}{Real\left(f_{\sigma }\right)}=-54.6\cdot Imag\left(f_{\sigma }\right)/Real\left(f_{\sigma }\right)$ vs. Al conductivity curves. 

The AE response shown in Figure 2a has the form of a relaxation response [13]; as the conductivity increases, the Rayleigh wave velocity decreases monotonically until it reaches a plateau, while the attenuation goes through a peak that occurs at a critical sheet conductivity $\sigma_{c}=v_{0}\left(\varepsilon_{0}+\varepsilon_{s}\right)$. According to Equations (7) and (8), the magnitude of the drop of the relative velocity change (i.e., the relative velocity shift associated with electrical shorting of the ZnO surface) is equal to K22: twice this value (about 0.0045) corresponds to the *K^2^* (about 1%) of the Rayleigh wave in c-ZnO half-space. If α/k is plotted vs. the relative velocity change Δvv0 (being $v_{0}$ the wave velocity along the bare ZnO surface) with the conductivity as the variable parameter, as shown in Figure 2b, the AE interaction assumes the form of a semi-ellipse centered at (−K24, 0), the angular position along the semiellipse corresponding to the conductivity of the overlayer, in accordance with [16].

## 3. The Volume AE Effect

If the electrical conductivity of the piezoelectric medium undergoes an inhomogeneous in-depth variation (as in the case of UV radiation absorption, to cite an example), then the induced AE effect is not a surface but a volume effect (VAE effect). An exponentially decaying spatial distribution of the ZnO conductivity σ(y)=σ0·e−yδsd was assumed, where *σ*_0_ is the surface conductivity, δ_sd_ is the *skin depth*, the distance into the ZnO material at which the conductivity has dropped by a factor of 1/e. The method adopted to account for the inhomogeneous conductivity distribution in the ZnO is to consider it as a stratified material with characteristics slowly varying over the layers. The wave propagating medium consists of a stacking sequence of 60 ZnO layers of equal thickness (δ = 40 nm), each layer having a different complex permittivity value: the imaginary part of the complex permittivity is frequency dependent according to $j\cdot \frac{h_{i}\sigma }{\omega \varepsilon_{0}}$ where the electrical conductivity σ is the sweep parameter and *h_i_* is a dimensionless weighting factor that accounts for the in-depth exponentially degrading conductivity according to the expression hi=e−(i−1)δδsd· (1+e−δδsd2), for *i* = 1 to 60, and δ the thickness of the layers, as shown in Figure 3. The stacking sequence of 60 ZnO layers covers a ZnO half-space, 7·λ thick, having real permittivity. Figure 3 shows, as an example, the column plot of $e^{-y/\delta_{sd}}$ vs. depth curve for δ_sd_ = 100 nm: the black points show the mean value *h_i_* of the ordinates, which corresponds to two successive layers of width 40 nm.

### Simulation Methodology and Results

A 2D FEM study was performed by using Comsol 5.6 software to evaluate the SAW velocity changes of the Rayleigh wave under UV illumination: Figure 4a,b shows the schematic of the SAW propagation path illuminated by UV light and the unit cell configuration. The cell consists of an insulating ZnO domain (64 µm thick and λ = 10 µm wide) having real and fixed permittivity; a discretized domain of 60 thin ZnO layers (with thickness δ = 40 nm) having frequency-dependent complex permittivity that varies over the layers; the air domain (2λ thick). A sweep parameter study was performed to calculate the mode eigenfrequency $f_{\sigma }$ at different ZnO conductivities (from 10^−7^ to 10^16^ S/m) and for different δ_sd_ values (from 100 to 500 nm). The δ_sd_ values were chosen among those reported in the available literature: for incident UV light at 365 nm, the penetration depth in ZnO ranges from 80 to a few a hundred nanometers [17,18,19,20,21,22] since it is strongly related to the structural properties of the photo-conducting material.

Figure 5a,b shows the relative velocity change $\Delta v/v_{0}$ and the propagation loss $\alpha =-54.6\cdot Imag\left(f_{\sigma }\right)/Real\left(f_{\sigma }\right)$ vs. the ZnO conductivity curves of the Rayleigh wave traveling along the surface of a ZnO half-space, being δ_sd_ the running parameter. The black curves in Figure 5 represent the case of ZnO half-space covered by a thin Al layer, as described in the previous paragraph.

The relative velocity change and propagation loss curves of Figure 5a,b have the form of a *double*-relaxation response as opposed to the black-colored curves which refer to the *surface* conductivity changes (already shown in Figure 2). The different behavior of the two AE effects is particularly evident if α/k is plotted vs. Δvv0, with σ as the variable parameter, as shown in Figure 6, being k = 2π/λ the acoustic wavevector; the curves correspond to different values of δ_sd_ (which varies in the range from 100 to 500 nm); the black curve represents the case of ZnO half-space covered by a thin Al layer. The curves can be resolved into two semi-ellipses, contrary to the single arc found in the previous case; the overlapping of the semi-ellipses increases with increasing the δ_sd_. The propagation loss $\alpha /k=-54.6\cdot Imag\left(f_{\sigma }\right)/\left(k\cdot Real\left(f_{\sigma }\right)\right)$ referred to as the *surface* AE effect (the black curve in Figure 5b) reaches an amplitude larger than that referred to as the *volume* AE effect since the imaginary part of the resonant frequency (0.58 MHz) is larger than that referred to the volume AE effect (from 0.48 to 0.55 MHz); with increasing the δ_sd_ value, the maximum value of Imag(f) increases.

The diagram in Figure 6 represents a series of two interlinked semi-ellipses indicating two relaxation mechanisms separated by conductivity. These results may be explained as due to two different dynamics, each relaxing at a different critical conductivity: for conductivity values corresponding to the first plateau, only the surface layers of the stacked region become conductive and contribute to the AE effect in a manner quite like the *surface* AE effect. With a further increase of the UV intensity, also deeper layers (placed in between the conductive surface layers and the insulating ZnO half-space) take part in the AE effect, as will be further discussed in the following paragraph.

## 4. Discussions

The Rayleigh wave phase velocity vs. σ curves refer to a non-homogeneous conductivity distribution in the ZnO half-space and show some differences with respect to the curve referring to the surface conductivity change: (1) the curves show two plateaus that are less pronounced for increasing δ_sd_ values; (2) the amplitude of the first velocity drop is similar to that of the *surface* AE effect; (3) the first drop is moved toward abscissa values lower than those referred to the *surface* AE effect; and (4) the magnitude of the total velocity drop of the *volume* AE effect exceeds that of the *surface* AE effect. 

The propagation loss vs. σ curves referred to the *volume* AE effect show some differences respect to the *surface* AE effect: (1) the curves show two peaks of different amplitude; (2) the dominant peak is downshifted respect to the peak of the *surface* AE effect; (3) the second small peak moves toward the larger one with increasing δ_sd_ and quite merges to the first peak at δ_sd_ = 500 nm; (4) the amplitude of the large peaks increases with increasing the δ_sd_. 

The anomalous behavior of the *volume* AE effect appears when the modulation of the conductivity becomes strong inside deep layers of the stacked region. With increasing values of the parameter σ, the number of layers involved in the conductivity modulation increases. Figure 7a–d shows the velocity and propagation loss vs. conductivity curves for δ_sd_ = 200 nm, and the electric potential Ф shape, for some conductivity values, as an example.

For very low conductivity values (Figure 7b), the ZnO is insulating: the SAW characteristics are unaffected by σ and the electric potential Ф at the ZnO free surface is non-null and continuous in the adjacent medium (air); Ф tends asymptotically to zero in both the ZnO and air domains. If the conductivity is increased, a screening effect occurs, causing a shorting of the electric field that results in power loss and piezoelectric stiffening. For conductivity values corresponding to the first plateau (Figure 7c), the system consists of an insulating ZnO half-space covered by a thin ZnO over-layer of low resistivity: the electric potential in the over-layer is constant and continuous with that in the air (overlying domain) and in the underlying half-space. With a further increase of the conductivity, the surface layers become conductive (the electrical potential is zero there), while the conductivity of the deep layers of the stacked region starts to increase. Finally, as shown in Figure 7d, the shielding effect is observed in the total stacked region, which is now conductive: Ф is null in air and in the stacked region, and it re-assumes its well-known profile inside the piezoelectric ZnO half-space, where it falls off exponentially with distance from the ZnO surface. 

Another effect that certainly contributes to the *volume* AE response of the Rayleigh wave in stratified ZnO half-space is that linked to the dispersion of the phase velocity: when the first layers of the stacked region become conductive, the SAW travels in a sort of *bi-layered structure* consisting of a ZnO half-space covered by a *conductive* ZnO film; the propagating medium becomes a “*slow on fast*” structure where the *slow* layer covers a *fast* substrate. The investigation of the profile of the electric potential inside the ZnO half-space suggests that, in addition to the velocity and attenuation changes deriving from the perturbations of the electric conductivity, there is also a dispersion effect due to the increasing number of layers that become conductive by increasing the σ parameter value. The presence of the *slow* layers alters the mechanical and electrical boundary conditions for the SAW propagation; the wave velocity and attenuation are affected by the acoustic energy radiation into the conductive lossy overlayer. As the thickness of the conductive overlayer increases, the wave velocity and propagation loss become sensitive to the mass loading from the conductive layer at the top surface. Once the whole stacked layer has become conductive, the wave velocity stabilizes even in presence of further conductivity increase. We can make the hypothesis that the dispersion effect due to the mass loading contributes to the *total* drop of the wave velocity (which is twice as large as the drop caused by the *surface* AE effect) and the tilting of the attenuation peak. 

Although the explanations provided might be physically sound, the question of other possible reasons contributing to the *volume* AE effect remains open. 

## 5. Conclusions

The acousto-electric (AE) effect associated with the propagation of the Rayleigh wave in ZnO half-space was investigated under the hypothesis that the ZnO conductivity change extends through a stacked region of the piezoelectric substrate according to an exponentially decaying conductivity profile. This hypothetical condition may recall the UV light absorption in ZnO which is a wide band gap semiconductor: it is expected that the exponential law of light adsorption into the ZnO is equal to the law that defines the spatial distribution of the ZnO photo-conductivity features and that the conductivity change is related to the incident UV power source. 

The changes in wave velocity *v* and propagation loss α induced by the in-depth, inhomogeneous changes in the ZnO electrical conductivity were calculated by a 2D FEM study for different values of the decay constant (from 100 to 500 nm). Plotting the wave propagation loss vs. the wave relative velocity change, α/k vs. Δ*v*/*v*_o_, with the ZnO electrical conductivity as a variable parameter, gives the representation of the double-relaxation response as opposed to the single-relaxation response which characterizes the well-known AE effect due to *surface* conductivity changes onto piezoelectric media. The calculated *volume* AE response showed that initially, it is only the more superficial layers of the stacked region that become conductive and therefore contribute to the AE effect in a manner quite similar to the *surface* AE effect: the configuration can be schematized as that of a surface layer (with variable conductivity) coated on a piezoelectric half-space. Subsequently, with the further increase of the conductivity (with the further increase in the absorbed UV power), the more superficial layers of the stacked region remain conductive while the underlying layers begin to become conductors and contribute to the AE effect: the configuration can be schematized as that of a conductive layer coated on a layer (with variable conductivity)/piezoelectric half-space.

The investigation of the electric potential profile inside the ZnO half-space, for different σ and δ_sd_, suggests that, other than the velocity and attenuation changes arising from perturbed electrical conditions, also a velocity dispersion effect takes place due to the increased thickness of the ZnO conductive-overlayer while increasing the conductivity.

It is worth saying that the accuracy of the calculated results depends on the number N and thickness δ of the ZnO layers in the stacked region, and on δ_sd_: by using N from 10 to 60, δ from 100 to 300 nm, and δ_sd_ from 240 to 1000 nm the double-relaxation effect is always clearly visible; the choice of δ = 40 nm, N = 60 represents a good compromise if a δ_sd_ value ranging from 100 to 500 nm is selected.

As to the author’s knowledge, the double-relaxation AE effect has not yet been predicted theoretically or observed experimentally (there are no practical experiments published that validate a theory up-to-now unknown, but it cannot be excluded that anomalous SAW UV sensor responses have been experimentally observed). The available scientific literature reports many examples of experimental measurements of the UV sensing performances of SAW sensors based on layers of ZnO as thick as the UV penetration depth: this condition satisfies the assumption that the entire ZnO layer undergoes a homogeneous conductivity change under UV illumination but makes it impossible even to guess the presence of the double-relaxation effect. Reference [23] clearly mentions the intentional choice of the ZnO layer thickness (200 nm), which is close to the experimentally estimated penetration depth of UV light (around 183 nm): the ZnO/LiNbO_3_ SAW sensor showed a frequency shift of 170 kHz under 40 mW/cm^2^ UV power illumination. In reference [24], a ZnO layer as thin as 71 nm is used as a UV sensing element: the authors say that this choice is suitable to avoid the perturbation of the wave propagation in the LiNbO_3_ substrate from the ZnO mass loading. In reference [25], the ZnO(250 nm)/128°yx-LiNbO_3_ showed a sensitivity of 6000 ppm/(μW/cm^2^) in a wide UV power range (from 0.010 to 40 mW/cm^2^): despite the wide UV power range, the double-relaxation phenomenon is not observed, probably due to the small ZnO layer thickness. In reference [26], however, the UV-induced frequency shifts of the Sezawa wave in ZnO (3.23 µm)/Si-based SAW oscillator were measured in a wide range of UV light intensities: the frequency shift vs. UV power curve exhibits two sensitivities which are 8.12 ppm/(μW/cm^2^) in the low power region (up to about 50 µW/cm^2^) and 1.62 ppm/(μW/cm^2^) in the high power region (from about 50 to 551 µW/cm^2^); unfortunately, the authors did not study the effect of the UV power on the wave propagation loss. The existence of two slopes in the frequency shift vs. the UV power source curve is attributed by the authors to the saturation of photogenerated carriers. It would have been useful to have some more measurements of both frequency shift and propagation loss over a wider range of UV powers (>551 µW/cm^2^) to verify if the hypothesis of an approaching plateau due to a double-relaxation phenomenon is realistic.

Studies are currently under way since the double-relaxation AE effect can be expected also in other coupling configurations, such as piezoelectric film onto non-piezoelectric half-space, which can be used to implement SAW UV sensors suitable for top and bottom UV illumination [27]. 

## Figures and Tables

**Figure 1 sensors-23-01422-f001:**
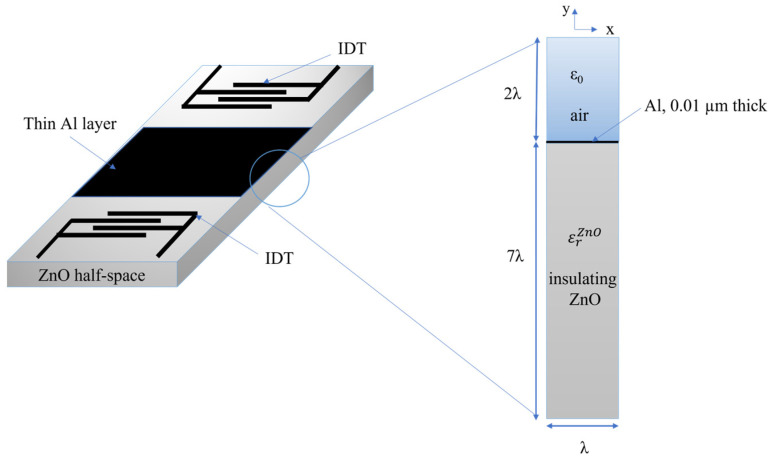
Schematic of the SAW delay line together with the detail of the unit cell (not in scale) including the ZnO domain (7λ thick), the Al domain (0.01 µm thick), and the air domain (2λ thick).

**Figure 2 sensors-23-01422-f002:**
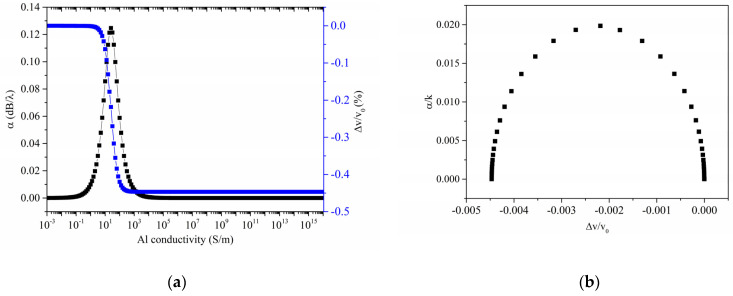
(**a**) The phase velocity and propagation loss vs. Al conductivity curves; (**b**) the parametric representation of the AE response, the α/k vs. Δvv0 curve with the conductivity as parameter, being k = 2π/λ the wavevector.

**Figure 3 sensors-23-01422-f003:**
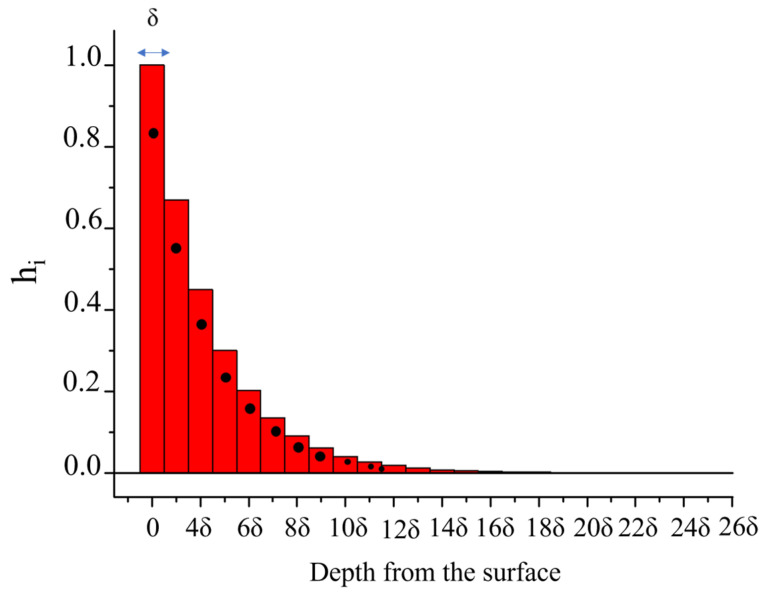
The column plot of the $e^{-y/\delta_{sd}}$ vs. depth curve, assuming δ_sd_ = 100 nm: the black points show *h_i_*, the mean value of the ordinates corresponding to two successive layers of width 40 nm.

**Figure 4 sensors-23-01422-f004:**
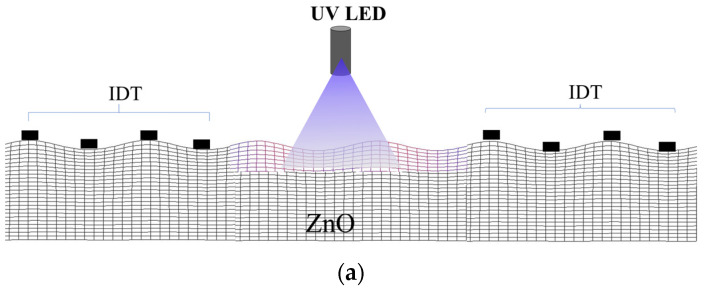
(**a**) Cross-section of the SAW delay line illuminated by UV radiation; (**b**) FEM unit cell of the ZnO half-space (partially discretized) and air (the picture is not in scale).

**Figure 5 sensors-23-01422-f005:**
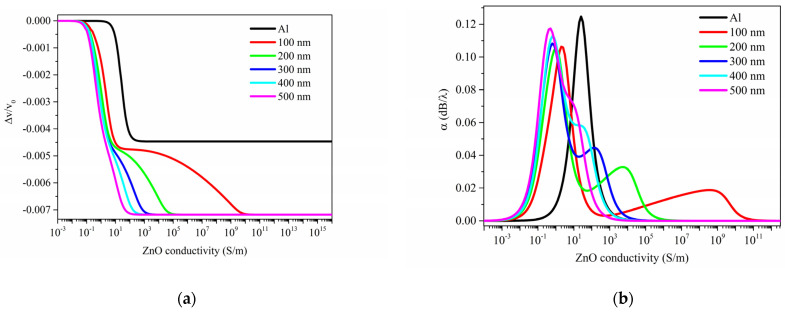
(**a**) Relative phase velocity change Δvv0  and (**b**) propagation loss α vs. ZnO conductivity curves for different values of the UV penetration depth $\delta_{sd}$ (from 100 to 500 nm).

**Figure 6 sensors-23-01422-f006:**
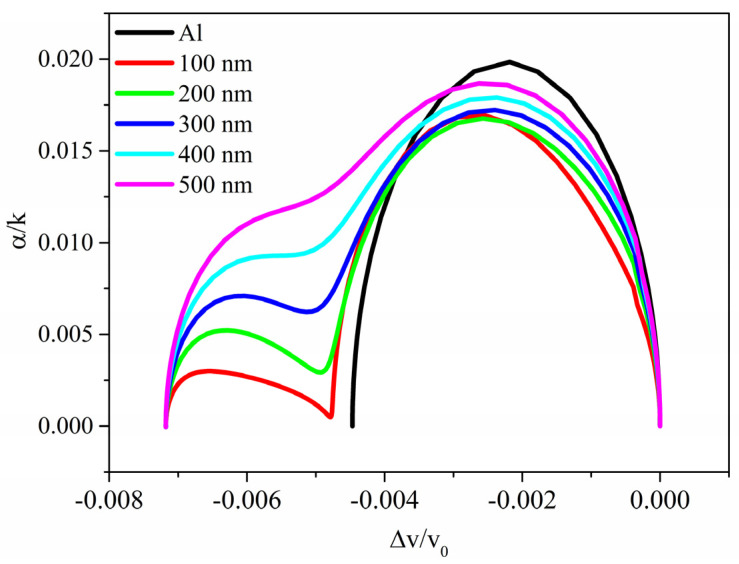
The α/k vs. Δvv0 curves, with σ as the variable parameter for different values of δ_sd_ (which varies in the range from 100 to 500 nm); the black curve represents the case of ZnO half-space covered by a thin Al layer.

**Figure 7 sensors-23-01422-f007:**
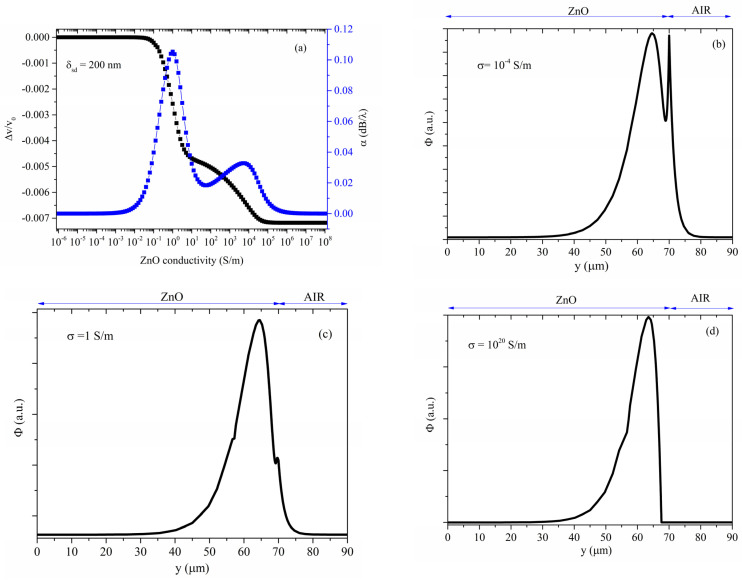
(**a**) Velocity and propagation loss vs. the ZnO conductivity curves; the electric potential profile for σ equal to (**b**) 0.0001, (**c**) 1, and (**d**) 10^20^ S/m; all the plots refer to δ_sd_ = 200 nm.

## Data Availability

The data can be found at https://github.com/matieber/mobilebattprep (accessed on 30 December 2022), which is under construction and will be rapidly built.

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
