# Peer review of "Acoustoelectric Effect for Rayleigh Wave in ZnO Produced by an Inhomogeneous In-Depth Electrical Conductivity Profile"

_sensors, 2023, doi:10.3390/s23031422_

Round 1
Reviewer 1 Report
1 comment.
Authors should shorten the introduction by removing well-known knowledge about piezoelectric materials.
2 comment.
The authors should more clearly state the geometry of the problem. In particular, for this purpose it is desirable to add a figure with the geometry of the problem.
3 comment
On the figures 7 b-d there are no designations of the axes.
Author Response
please consider the file uploaded

Reviewer 2 Report
This manuscript offers a simulation of the acoustic-electric effect (related to the piezoelectric property) of the Rayleigh wave on the ZnO surface. Among the notable conclusions is the novel double-relaxation effect reported. The piezoelectric response can be utilized for many sensing applications. Therefore, the simulation provided here is attractive to the journal’s readership.
The manuscript is primarily well-written and well-argued and can be published after minor revisions:
The first point is that it is not easily well understood from the text why should ZnO’s conductivity vary with depth from the surface. The observation of the double relaxation effect is contingent upon ZnO having an exponentially decaying conductivity profile from the surface. The validity of this assumption must be further discussed in the manuscript. From a materials science viewpoint, conductivity in ZnO can arise from intentional dopants [https://doi.org/10.1063/1.4719977], or unintentional defects [https://doi.org/10.1103/PhysRevLett.85.1012]. These dopants and defects are generally easier to form at ZnO’s surface rather than deeper away from the surface [https://doi.org/10.1016/S0039-6028(03)00273-5]. Otherwise, pristine and perfect ZnO is an insulator.
Second, some of the labels in Figure 7 are missing.
Finally, in the abstract’s last word, the author probably means “predicted” instead of “observed.”
Author Response
please consider the file uploaded

Reviewer 3 Report
The manuscript reports a double-relaxation response of the acoustic-electric effect of Rayleigh wave in ZnO. It is found that the double-relaxation is due to the surface conductivity activities of piezoelectric material. The topic discussed is relatively novel and may impact and benefits the related research field. After reading the article carefully, we have concluded that the article has shown effort. However, it still needs some changes before it is ready to be published.
1. First and foremost, it is recommended to cite articles from recent publications (2020-2022) to justify that the work is at an ascending level.
2. Please cite reputable theoretical articles or books for each equation.
3. All the figures are not of decent quality, please increase the resolution for all the figures according to the guidelines provided by the publisher.
4 There are some grammatical, formatting and word spelling errors. Please review the article carefully.
5. In page 8, the author shows the double-relaxation, and simply claims “………….due to different dynamics, each relaxing at a different critical conductivity.” Such an explanation is insufficient, please explain in more detail and support the statement with references. In addition, in Figure 8, please explain why the Al curve is above all the others.
6. In page 6, the author mentioned “……, the system consists in ZnO insulting half-space with a thin iso-potential low-conductivity ZnO overlayer where Ф is constant.” Please discuss the iso-potential mechanism.
Author Response
please consider the file uploaded

Reviewer 4 Report
The manuscript describes FEM simulation studies on the acoustoelastic scattering on a ZnO/Al material compound.
The results are clearly presented, however, the description which parameters are used in equation 8 could be improved. Also, the references 3~ 17 are more than ten years old, to my opinion they are rather inappropriate for this research, while an important paper on the "valley acoustic-effect" is not cited: doi:10.1103/PhysRevLett.122.256801
Author Response
please consider the file uploaded
